# A kinesin-13 family kinesin in *Trypanosoma brucei* regulates cytokinesis and cytoskeleton morphogenesis by promoting microtubule bundling

Huiqing Hu[�], Yasuhiro Kurasawa[☉], Qing Zhou, Ziyin Li[iD]*

Department of Microbiology and Molecular Genetics, McGovern Medical School, University of Texas Health Science Center at Houston, Houston, Texas, United States of America

☉ These authors contributed equally to this work.
* Ziyin.Li@uth.tmc.edu

**Data Availability Statement:** All data are included in the manuscript and its Supporting information files.

## Abstract

The early branching eukaryote *Trypanosoma brucei* divides uni-directionally along the longitudinal cell axis from the cell anterior toward the cell posterior, and the cleavage furrow ingresses along the cell division plane between the new and the old flagella of a dividing bi-flagellated cell. Regulation of cytokinesis in *T. brucei* involves actomyosin-independent machineries and trypanosome-specific signaling pathways, but the molecular mechanisms underlying cell division plane positioning remain poorly understood. Here we report a kinesin-13 family protein, KIN13-5, that functions downstream of FPRC in the cytokinesis regulatory pathway and determines cell division plane placement. KIN13-5 localizes to multiple cytoskeletal structures, interacts with FPRC, and depends on FPRC for localization to the site of cytokinesis initiation. Knockdown of KIN13-5 causes loss of microtubule bundling at both ends of the cell division plane, leading to mis-placement of the cleavage furrow and unequal cytokinesis, and at the posterior cell tip, causing the formation of a blunt posterior. *In vitro* biochemical assays demonstrate that KIN13-5 bundles microtubules, providing mechanistic insights into the role of KIN13-5 in cytokinesis and posterior morphogenesis. Altogether, KIN13-5 promotes microtubule bundle formation to ensure cleavage furrow placement and to maintain posterior cytoskeleton morphology in *T. brucei*.

## Author summary

Cytokinesis is the final stage of cell division, and in the early branching eukaryote *Trypanosoma brucei* cytokinesis occurs uni-directionally along the cell division plane that is placed between the new and the old flagella of a bi-flagellated cell. Regulation of cytokinesis in *T. brucei* depends on actomyosin-independent machineries and trypanosome-specific signaling pathways, but the mechanisms underlying cell division plane placement remain enigmatic. In this work, we report a kinesin-13 family protein named KIN13-5 that determines cell division plane positioning. KIN13-5 localizes to the plus ends of

**Funding:** This work was supported by the National Institutes of Health R01 grants AI101437 and AI118736 to Z.L. The funders do not play any role in the study design, data collection and analysis, decision to publish, or preparation of the manuscript.

**Competing interests:** The authors have declared that no competing interests exist.

cytoskeletal microtubules at multiple locations and functions downstream of the cytokinesis regulator FPRC in the cytokinesis regulatory pathway. Knockdown of KIN13-5 by RNAi disrupts microtubule bundling, causing mis-placement of the cleavage furrow and asymmetrical cytokinesis and the formation of a blunt posterior. Purified recombinant KIN13-5 bundles microtubules *in vitro*, which provides biochemical evidence for the molecular function of KIN13-5 in cytokinesis and posterior morphogenesis. Together, through its microtubule bundling activity KIN13-5 plays an essential function in cleavage furrow placement and posterior cytoskeleton morphogenesis in *T. brucei*.

## Introduction

Cytokinesis is the final step of cell division in all living organisms, and is regulated by distinct mechanisms and molecular machineries in bacteria, fungi, animals, and plants. Bacteria use the FtsZ contractile ring located at the cell division plane for cell division, fungi and animals employ the actomyosin contractile ring to divide the cell, and plants use membrane fusion and cell wall construction along the cell division plane for cytokinesis [1,2]. The actomyosin-dependent cytokinesis regulatory pathway in fungi and animals involves two evolutionarily conserved kinases, the Polo-like kinase and the Aurora B kinase, which cooperate to phosphorylate the Centralspindlin complex located at the central spindle for activation of the small GTPase RhoA that further promotes the assembly of the actomyosin contractile ring complex [3]. Actomyosin-independent cytokinesis regulatory pathways have been reported to operate in early divergent parasitic protozoa, including *Trypanosoma brucei* and *Giardia lamblia*, which divide along the longitudinal cell axis and depend on some species-specific regulatory proteins and certain evolutionarily conserved regulatory proteins, such as the Polo-like kinase and the Aurora B kinase [4,5]. However, the mechanisms underlying the actomyosin-independent cytokinesis in *T. brucei*, *G. lamblia*, and other protozoa remain poorly understood.

*T. brucei* is a flagellated, unicellular protozoan causing African sleeping sickness, and the regulation of its cell division has many unusual features. First, the cell division plane in *T. brucei* is not determined by the position of the central spindle as in animals [2], but rather is correlated with the length of the newly assembled flagellum and its associated cytoskeletal structure termed the flagellum attachment zone (FAZ) [6,7]. Secondly, a cell division fold is formed, through membrane invagination and cytoskeleton remodeling, along the cell division plane prior to the initiation of cytokinesis [8]. Thirdly, cytokinesis cleavage furrow ingresses unidirectionally along the cell division fold from the anterior tip of the new-flagellum daughter (NFD) cell toward the nascent posterior of the old-flagellum daughter (OFD) cell [8,9]. Finally, cleavage furrow ingression and final separation of the two daughter cells require flagellar motility [10]. These unusual features in cytokinesis make *T. brucei* an excellent model organism to study the mechanism and the regulation of actomyosin-independent cytokinesis in early branching eukaryotes.

The regulatory pathway controlling cytokinesis in *T. brucei* comprises evolutionarily conserved regulators, such as the Polo-like kinase homolog TbPLK, the Aurora B kinase homolog TbAUK1, and the katanin60-katanin80 complex, and trypanosome-specific regulators, including the kinetoplastid-specific protein phosphatase KPP1, the orphan kinesin KLIF, and other proteins (CIF1, CIF2, CIF3, CIF4, FPRC, and FRW1) with diverse structure motifs [9,11–23]. These regulatory proteins act in concert at the anterior tip of the NFD cell and the cleavage furrow to promote the initiation and/or the completion of cytokinesis, and the order of actions for these regulators has been determined. CIF4 and FPRC appear to act at the very upstream of

the cytokinesis regulatory pathway in *T. brucei* by recruiting CIF1 and many CIF1-interacting partner proteins to the anterior tip of the NFD cell [22]. CIF1 likely functions as a master regulator of cytokinesis, interacting with all of the known cytokinesis regulatory proteins and recruiting all of its interacting proteins except CIF4 and FPRC, whereas some CIF1-interacting proteins, including TbPLK, KPP1, CIF2, and CIF3, exert a feedback control on CIF1 by maintaining its stability or localization [17,19–22,24]. Additional cytokinesis regulatory proteins that function in this pathway remain to be identified and characterized for a comprehensive understanding of the signaling cascade governing the unusual mode of cytokinesis.

Cytokinesis in *T. brucei* requires faithful duplication and separation of the cytoskeleton, which is defined by an array of subpellicular microtubules underlying the plasma membrane [25]. The subpellicular microtubule array is polarized, with the microtubule plus-ends directed toward the posterior tip of the cell. During the cell division cycle, the microtubule-based cytoskeleton in *T. brucei* duplicates in a semi-conservative manner [25], and it undergoes extensive remodeling and morphogenesis, especially at the anterior tip of the NFD cell and the nascent posterior of the OFD cell of dividing cells [26]. Proper organization/remodeling of the subpellicular microtubule array at the nascent posterior of the OFD cell is critical for cytokinesis completion, and it is regulated by the orphan kinesin KLIF that bundles microtubules at the nascent posterior [27,28] and by several cytoskeleton-associated proteins that maintain KLIF localization at the cleavage furrow for the latter to bundle microtubules at the nascent posterior [27]. The posterior tip of trypanosome cells is maintained as a tapered shape, and the plus-ends of the subpellicular microtubules at the posterior cell tip are bound by several microtubule plus end-binding proteins, including XMAP215 and EB1 [8, 29]. Maintenance of a tapered shape at the cell posterior requires CRK2-mediated regulation of microtubule dynamics [30] and PAVE1-mediated stabilization of the microtubule array [31].

Here, we report the essential role of KIN13-5, a putative kinesin-13 family protein in *T. brucei*, in regulating posterior morphogenesis and cytokinesis by promoting microtubule bundling. KIN13-5 is an interacting partner of the cytokinesis regulatory protein FPRC and localizes to the posterior cell tip, the distal tip of the new and the old FAZ filaments, the nascent posterior of the OFD cell, and the basal body. Our work discovered the roles of KIN13-5 in maintaining the tapered shape at the posterior cell tip and placing the cell division plane or cleavage furrow for an equal cytokinesis through its microtubule-bundling activity. These findings highlight the essential involvement of a kinesin protein in regulating cell division plane positioning to ensure faithful cell division in *T. brucei*.

## Results

### Identification of KIN13-5 as an interacting partner of the cytokinesis regulator FPRC

To identify new regulators of cytokinesis that function in the CIF1-mediated cytokinesis signaling pathway, we recently performed BioID using CIF1 and FPRC as bait and identified several known cytokinesis regulators, a cohort of cytoskeleton-associated proteins, and several new FAZ tip-localized proteins, whose functions remain unknown [27] (Fig 1A). One of the new FAZ tip-localized proteins is KIN13-5, a kinesin-13 family protein whose cellular function was not investigated previously [32,33]. To test whether KIN13-5 interacts with CIF1 and/or FPRC *in vivo* in trypanosomes, we performed co-immunoprecipitation, using trypanosome cells co-expressing PTP-tagged FPRC and 3HA-tagged KIN13-5 or cells expressing 3HA-tagged KIN13-5. Immunoprecipitation of FPRC-PTP was able to pull down KIN13-5-3HA from trypanosome cell lysate (Fig 1B), but immunoprecipitation of KIN13-5-3HA did not pull down CIF1 (S1A Fig), suggesting that KIN13-5 interacts with FPRC, but not CIF1. We next

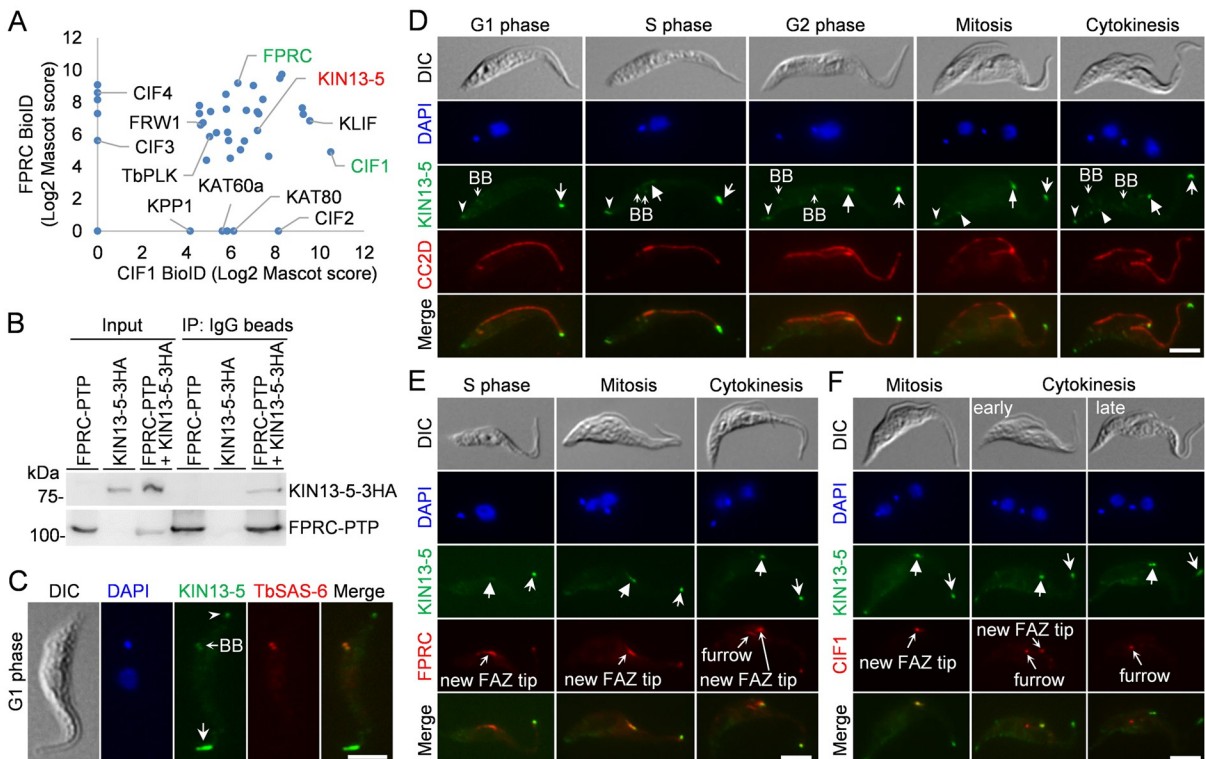

**Fig 1. Identification of KIN13-5 as an interacting partner of FPRC and determination of its subcellular localization.** (**A**). Identification of proximal proteins of FPRC and CIF1 by BioID. The BioID experiments were carried out previously [27]. Shown is the comparison of the proximal proteins of FPRC and CIF1. Known cytokinesis regulators, in addition to KIN13-5, were indicated. (**B**). Co-immunoprecipitation to examine the *in vivo* interaction between FPRC and KIN13-5, which were epitope-tagged with PTP or triple HA, respectively, in the same cell line. FPRC-PTP was precipitated with IgG beads, and detected by anti-Protein A antibody. The co-immunoprecipitated KIN13-5-3HA was detected by anti-HA antibody. (**C**). Localization of KIN13-5, which was endogenously tagged with a C-terminal triple HA epitope, in G1 cells. TbSAS-6 detects the basal body. Open arrowhead indicates KIN13-5-3HA at the posterior tip, whereas the arrow indicates KIN13-5-3HA at the FAZ tip. BB, basal body. Scale bar: 5 μm. (**D**). Localization of KIN13-5-3HA during the cell cycle of the procyclic form of *T. brucei*. CC2D labels the FAZ. Open arrowheads indicate KIN13-5-3HA at the posterior tip, open arrows indicate KIN13-5-3HA at the FAZ tip (G1 phase) or the old FAZ tip (S phase to cytokinesis), solid arrows indicate KIN13-5-3HA at the new FAZ tip (S phase to cytokinesis), and solid arrowheads indicate KIN13-5-3HA at the nascent posterior of the OFD cell (mitosis and cytokinesis). BB, basal body. Scale bar: 5 μm. (**E**). Co-localization of KIN13-5 with FPRC at the new and old FAZ tips. Cells co-expressing KIN13-5-3HA and FPRC-PTP from their respective endogenous locus were co-immunostained with FITC-conjugated anti-HA antibody and anti-Protein A antibody. Solid arrows indicate KIN13-5-3HA at the new FAZ tip, whereas open arrows indicate KIN13-5-3HA at the old FAZ tip. Scale bar: 5 μm. (**F**). Co-localization of KIN13-5 and CIF1 at the new FAZ tip. Endogenous KIN13-5 was tagged with a C-terminal triple HA epitope and detected by FITC-conjugated anti-HA antibody. CIF1 was detected by polyclonal anti-CIF1 antibody. The solid arrows indicate KIN13-5-3HA at the new FAZ tip, whereas the open arrows indicate KIN13-5-3HA at the old FAZ tip. Scale bar: 5 μm.

examined the subcellular localization of KIN13-5 in procyclic trypanosomes by immunofluorescence microscopy using cells expressing endogenously 3HA-tagged KIN13-5. In G1-phase cells, KIN13-5 was found to localize to the basal body, the FAZ tip, and the posterior cell tip (Fig 1C and 1D). Starting from S phase of the cell cycle, KIN13-5 gradually emerged at the new FAZ tip, in addition to the basal body, the old FAZ tip, and the posterior cell tip (Fig 1D), and starting from mitosis until cytokinesis, KIN13-5 was additionally detected at the nascent posterior tip of the old-flagellum daughter cell (Fig 1D). At the new and old FAZ tips, but not the cleavage furrow, KIN13-5 co-localized with FPRC, and at the new FAZ tip, KIN13-5 co-localized with CIF1 (Fig 1E and 1F), consistent with the finding that KIN13-5 interacts with FPRC and localizes in close proximity to CIF1 at the new FAZ tip. The localization of 3HA-tagged KIN13-5 is consistent with that of mNeonGreen-tagged KIN13-5 reported recently [34].

## KIN13-5 is required for cytokinesis in the procyclic form of *T. brucei*

The localization of KIN13-5 to the new FAZ tip and the interaction of KIN13-5 with FPRC, a crucial cytokinesis regulator in *T. brucei* [22], suggest that KIN13-5 may play a role in cytokinesis in *T. brucei*. To study the cellular function of KIN13-5, we carried out RNAi to ablate KIN13-5 expression in procyclic trypanosomes. Western blotting demonstrated the depletion of KIN13-5, which was endogenously tagged with a C-terminal triple HA epitope, after RNAi induction for 24 hours (Fig 2A). This knockdown of KIN13-5 protein caused severe growth defects (Fig 2B). Quantitation of cells with different numbers of nuclei and kinetoplasts (the cell's mitochondrial DNA network) showed that knockdown of KIN13-5 resulted in the initial accumulation of bi-nucleated (2N2K, two nuclei and two kinetoplasts) cells and the subsequent accumulation of polyploid (xNyK, x>2, y≥1) cells (Fig 2C), suggesting that cytokinesis was impaired by KIN13-5 RNAi. To further characterize the cytokinesis defects caused by KIN13-5 depletion, we imaged the cells with light microscopy and scanning electron microscopy, and we observed that the number of dividing bi-nucleated cells with a visible cleavage furrow was gradually increased from ~32% to ~96% after RNAi induction for 48 hours (Fig 2D–2F), suggesting a defect in the completion of cytokinesis. Notably, ~94% of the dividing bi-nucleated cells appeared to divide in a significantly unequal manner, producing a smaller-sized NFD cell and an OFD cell that appeared to be longer than the OFD cell of the non-induced control (Fig 2D, 2E, 2G and 2H). These smaller-sized NFD cells had various cell size (Fig 2D, 2E and 2G) and contained a nucleus or no nucleus (Fig 2D and 2I). It should be noted that in wild-type procyclic trypanosomes, cytokinesis is not totally equal, as the NFD cell has shorter flagellum and slightly smaller size than the OFD cell and it undergoes remodeling in early G1 phase after cytokinesis [26]. However, the NFD cell of dividing KIN13-5 RNAi cells was significantly smaller and skinnier than the NFD cell of dividing control cells (Fig 2E). These results suggest that the fidelity of cytokinesis was affected by KIN13-5 RNAi, likely due to the mis-positioning of the cleavage furrow. The polyploid cells also possessed one or multiple visible cleavage furrows (Fig 2J and 2K), further confirming that KIN13-5 knockdown impaired the completion of cytokinesis.

## Knockdown of KIN13-5 causes mis-placement of the cell division plane

The unequal cytokinesis caused by KIN13-5 knockdown (Fig 2) prompted us to examine the placement of the cell division plane or the cleavage furrow. In wild-type trypanosome cells, a cell division fold was formed by membrane invagination along the longitudinal axis of bi-nucleated cells between the NFD cell and the OFD cell (Fig 3Aa). The cell division fold extended from the anterior cell tip of the NFD cell toward the nascent posterior of the OFD cell (Fig 3Aa), and the cleavage furrow was ingressing along the cell division fold in the same direction (Fig 3Ab and 3Ac, yellow arrows). At the final stage of cytokinesis, the two daughter cells were connected by a thin thread of cytoplasm, termed the cytoplasmic bridge, between the nascent posterior of the OFD cell and the ventral side of the NFD cell (Fig 3Ad). In KIN13-5 RNAi cells, a cell division fold was still formed prior to cytokinesis initiation, although the fold appeared to be wider than in wild-type cells (Fig 3Ae). Strikingly, the nascent posterior of the OFD cell was placed in close proximity to the existing posterior of the NFD cell, resulting in the extension of the cell division fold to the posterior end of the cell (Fig 3Ae). During cytokinesis progression in the KIN13-5 RNAi cells, the cleavage furrow was ingressing along the cell division fold toward the posterior end, and the nascent posterior and the existing posterior were placed next to each other (Fig 3Af and 3Ag, yellow arrows). At the final stage of cytokinesis, the two daughter cells were connected by a cytoplasmic bridge between the two posterior ends (Fig 3Ah). These results suggest that KIN13-5 RNAi impairs the placement of the cell

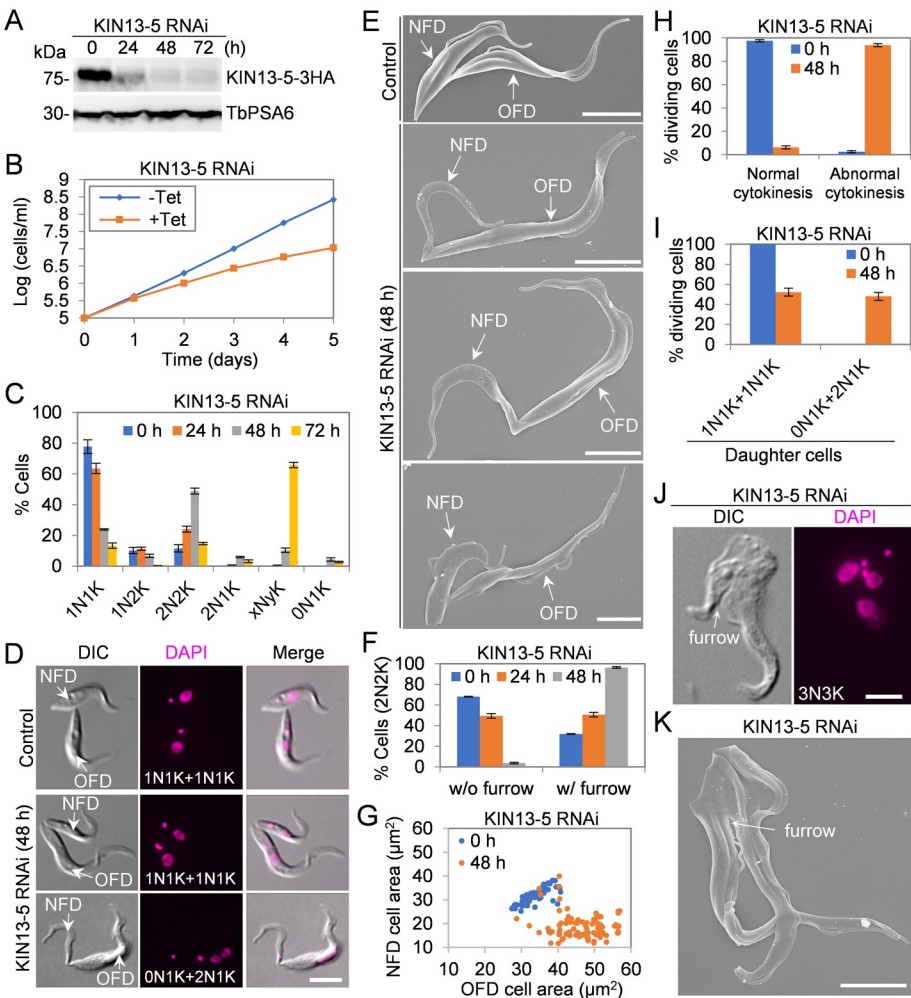

**Fig 2. Depletion of KIN13-5 by RNAi in procyclic trypanosomes caused cytokinesis defects.** (**A**). Western blotting to detect the KIN13-5 protein level during KIN13-5 RNAi induction. KIN13-5 was endogenously tagged with a C-terminal triple HA epitope in the KIN13-5 RNAi cell line. TbPSA6, which detects the α6 subunit of the 26S proteasome, serves as a loading control. (**B**). Knockdown of KIN13-5 by RNAi caused severe growth defects. (**C**). Knockdown of KIN13-5 by RNAi caused cytokinesis defects by increasing bi-nucleated cells (2N2K) and polyploid cells (xNyK, x>2, y≥1). Cells with different numbers of nuclei (N) and kinetoplasts (K) were counted and plotted. Error bars indicate S.D. from three independent experimental replicates. (**D**). Light microscopic analysis of dividing cells before and after KIN13-5 RNAi induction. NFD, new-flagellum daughter cell; OFD, old-flagellum daughter cell. Scale bar: 5 μm (**E**). Scanning electron microscopic analysis of dividing cells before and after KIN13-5 RNAi induction. Scale bar: 5 μm. (**F**). Quantitation of bi-nucleated cells with or without a cleavage furrow before and after KIN13-5 RNAi induction. Error bars indicate S.D. from three independent experimental replicates. (**G**). Measurement of the size of the NFD and OFD cells of the dividing bi-nucleated cells before and after KIN13-5 RNAi induction using ImageJ. (**H**). Quantitation of the dividing bi-nucleated cells for symmetrical (normal) division or asymmetrical (abnormal) division before and after KIN13-5 RNAi induction. Error bars indicate S.D. from three independent experimental replicates. (**I**). Quantitation of the dividing bi-nucleated cells for two 1N1K daughter cells or one 2N1K and one 0N1K daughter cells. Error bars indicate S.D. from three independent experimental replicates. (**J** and **K**). Multi-nucleated cells examined by light microscopy (**J**) and scanning electron microscopy (**K**). Scale bars: 5 μm.

division plane or cleavage furrow. Because of this defect, the OFD cell of the dividing KIN13-5 RNAi cells apparently had a longer posterior portion than the dividing control cells (Fig 3, compare 3Af-h with 3Ab-d).

Further, we used KLIF, an orphan kinesin localized to the ingressing cleavage furrow [17, 21], as a marker for the cleavage furrow or the cell division plane to examine the effect of

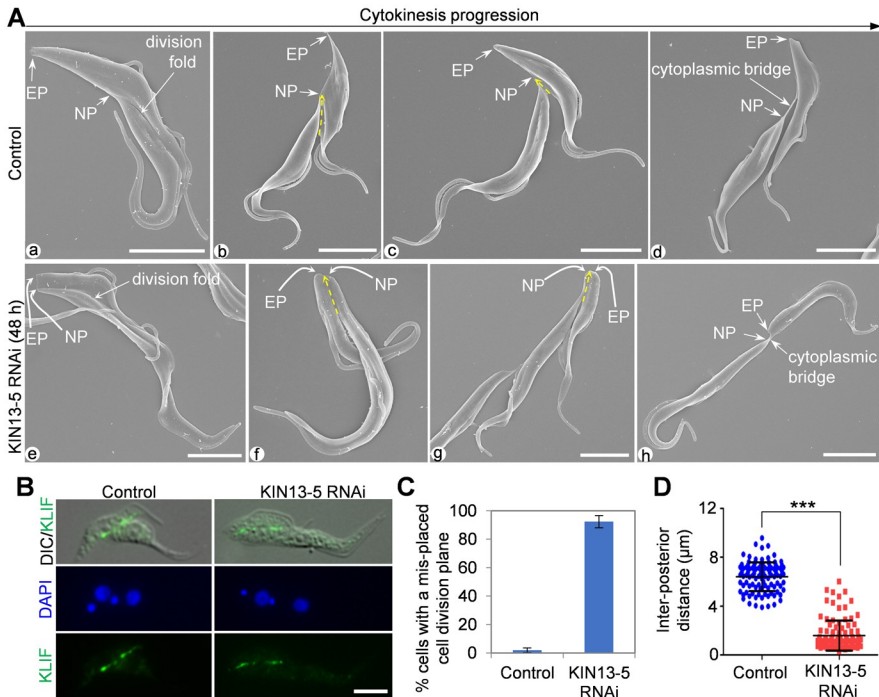

**Fig 3. Knockdown of KIN13-5 causes mis-placement of the cell division plane.** (**A**). Scanning electron microscopic analysis of the cell division plane in control and KIN13-5 RNAi cells. Yellow dash-line arrows indicate the cell division plane. EP, existing posterior; NP, nascent posterior. Scale bars: 5 μm. (**B**). Immunostaining of the cleavage furrow/cell division plane with the cleavage furrow marker protein KLIF. KLIF was endogenously tagged with a C-terminal triple HA epitope and detected by FITC-conjugated anti-HA antibody. Scale bar: 5 μm. (**C**). Quantitation of bi-nucleated cells with a mis-placed cell division plane. Error bars indicate S.D. from three independent experimental replicates. (**D**). Measurement of the inter-posterior distance in control and KIN13-5 RNAi cells using ImageJ. ***, $p < 0.001$.

KIN13-5 RNAi on the placement of the cleavage furrow or the cell division plane, and the results confirmed that the cleavage furrow/cell division plane was mis-positioned in ~92% of the dividing bi-nucleated cells after KIN13-5 RNAi for 48 hours (Fig 3B and 3C). Consequently, the distance between the nascent posterior and the existing posterior was significantly reduced after KIN13-5 RNAi (Fig 3D). Together, these observations demonstrated that knockdown of KIN13-5 causes mis-placement of the cell division plane or the cleavage furrow.

## Knockdown of KIN13-5 disrupts the morphology of the posterior cytoskeleton

The posterior tip of a trypanosome cell is comprised of the microtubule plus-ends of the sub-pellicular microtubule array and assumes a tapered shape [35]. In some KIN13-5 RNAi cells, we observed a blunt posterior end (Fig 2E), indicating that KIN13-5 may be involved in maintaining the morphology of the cell posterior. To characterize this potential morphology defect in detail, we examined the KIN13-5 RNAi cells under light microscope and scanning electron microscope (SEM), and we observed a blunt posterior end in the cells at different cell cycle stages, as determined by the number of kinetoplast and nucleus (for light microscopy) and the number of flagellum (for SEM) (Fig 4A). With the cells at all cell cycle stages combined, those with a blunt posterior end constituted ~30% of the total cell population after KIN13-5 RNAi induction for 48 hours (Fig 4B). Further, we performed immunostaining with the anti-HA antibody against the 3HA-tagged XMAP215, a microtubule plus-end-binding protein localized

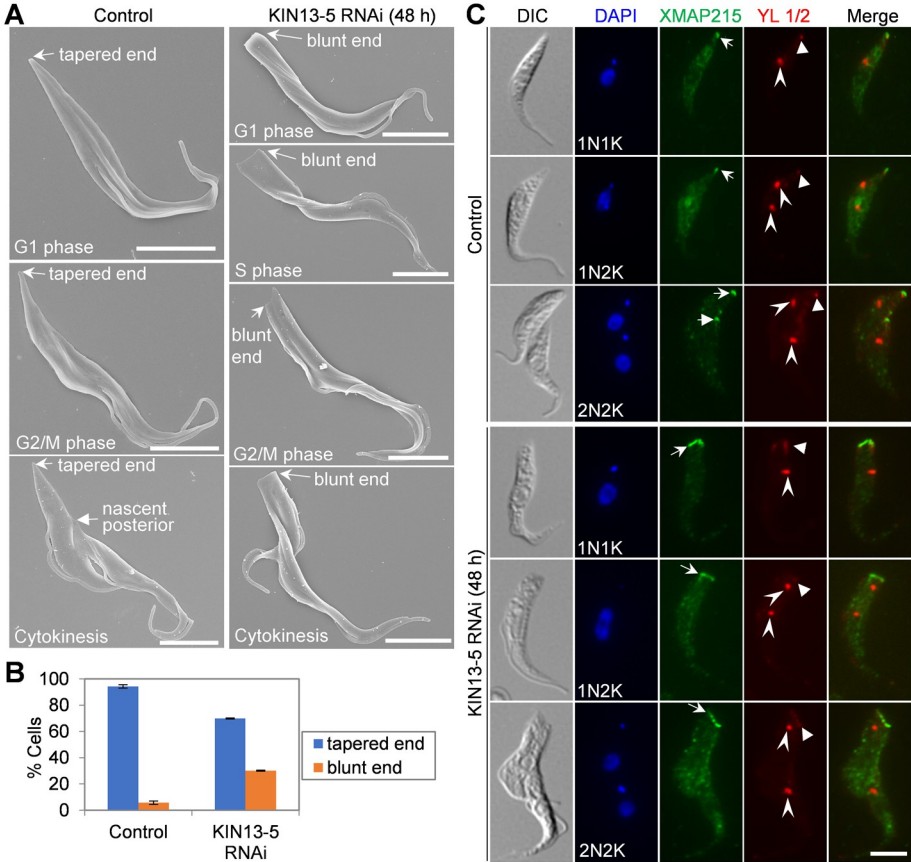

**Fig 4. KIN13-5 knockdown impairs posterior cytoskeleton morphogenesis.** (**A**). Scanning electron microscopic analysis of the posterior morphology in control and KIN13-5 RNAi cells. Cell cycle stages were determined based on the number and length of the flagellum as well as the cleavage furrow. Scale bars: 5 μm. (**B**). Quantitation of cells with different posterior shapes. Error bars represent S.D. from three independent experimental replicates. (**C**). Immunolabeling of the cell posterior with the microtubule plus-end-binding protein XMAP215. XMAP215 was endogenously tagged with a C-terminal triple HA epitope and detected with FITC-conjugated anti-HA antibody. Open arrows indicate XMAP215 signal at the posterior tip, solid arrows indicate XMAP215 at the nascent posterior, open arrowheads indicate basal body detected by the YL 1/2 antibody, and solid arrowheads indicate the newly assembled microtubules at the cell posterior. Scale bar: 5 μm.

to the posterior tip of trypanosome cells [8], to monitor the organization of the microtubule plus ends. In non-induced control cells, XMAP215 was detected as a round- or oval-shaped spot at the posterior of 1N1K and 1N2K cells (Fig 4C, open arrows) and at both the existing posterior and the nascent posterior of 2N2K cells (Fig 4C, open and solid arrows, respectively), indicating the convergence of the microtubule plus-ends at the posterior cell tip. In KIN13-5 RNAi cells, however, XMAP215 fluorescence signal assumed a bar shape at the blunt posterior of cells (Fig 4C, open arrows), suggesting that the plus-ends of the subpellicular microtubules were not bundled together in these cells.

Because the posterior portion of the OFD cell of the bi-nucleated cells appeared to be longer, we co-immunostained these cells with the YL 1/2 antibody, which labels newly assembled (tyrosinated) microtubules at the cell posterior [36, 37] and detects the RP2 protein at the mature basal body [38], to test whether the elongated posterior contains excessive newly-assembled microtubules, as observed in the elongated posterior of cells depleted of CRK2, CRK1, or some G1 cyclin genes [30, 39–41]. In control cells, the YL 1/2 antibody labeled the

basal body and the posterior tip, where new microtubules were assembled at the plus-ends of the subpellicular microtubules (Fig 4C, solid arrowheads). In KIN13-5 RNAi cells, the YL 1/2 fluorescence signal was still detected at the tapered posterior, similar to the XMAP215 fluorescence signal but with weaker intensity (Fig 4C, solid arrowheads). Notably, the posterior portion (from the posterior tip to the kinetoplast) of all the KIN13-5 RNAi cells examined were not intensively stained by YL 1/2 (Fig 4C), demonstrating that extensive microtubule assembly had not occurred at the posterior portion of the KIN13-5 RNAi cells. Thus, the longer posterior portion of the OFD cell of the dividing KIN13-5 RNAi cells was not due to microtubule extension, but rather was attributed to the mis-placement of the cleavage furrow, as demonstrated by scanning electron microscopic and immunofluorescence microscopic analyses (Fig 3). Altogether, these results suggest that KIN13-5 is required to maintain a tapered shape of the cell posterior by promoting subpellicular microtubule convergence at the cell posterior.

## RNAi of KIN13-5 impairs microtubule bundling at the cell posterior

We further investigated the effect of KIN13-5 RNAi on the organization of the subpellicular microtubules by electron microscopic analysis of detergent-extracted cytoskeletons. We first examined the microtubules at the posterior tip of 1N1K cells from both the non-induced control and the KIN13-5 RNAi-induced population. In the control 1N1K cells, the tips of microtubules at the cell posterior appeared to be converged together (Fig 5A, inset), which contributed to the formation of a tapered posterior end. Convergence of microtubule tips at the cell posterior was similarly observed in other cell types, including the cells that were just before cytokinesis initiation and were undergoing cytokinesis (Fig 5B–5D). In the KIN13-5 RNAi-induced 1N1K cells, however, the tips of microtubules at the cell posterior were widely spread out (Fig 5E, inset), which apparently contributed to the formation of a blunt posterior end. Similar arrangement of microtubule tips at the posterior cell end was observed in other cell types (Fig 5F–5H), consistent with the presence of a blunt posterior in all cell types observed by light microscopy and scanning electron microscopy (Fig 4).

We next focused on the microtubules at the anterior tip of the NFD cell and the nascent posterior of the OFD cell before cytokinesis initiation and during cytokinesis progression in control and KIN13-5 RNAi-induced cells. Prior to cytokinesis initiation in control cells, individual microtubules at the anterior tip of the NFD cell and at the nascent posterior of the OFD cell did not appear to form bundles (Fig 5B, insets), likely because at this cell cycle stage the NFD anterior tip and the OFD nascent posterior had not formed yet. In KIN13-5 RNAi cells, the microtubules at the NFD anterior tip and at the OFD nascent posterior, which was positioned close to the posterior of the NFD cell, also did not form bundles (Fig 5F, insets). In the control cells at early stages of cytokinesis with a short cleavage furrow, the microtubules at the NFD anterior tip, but not the microtubules at the OFD nascent posterior, appeared to form bundles (Fig 5C, insets). In the KIN13-5 RNAi-induced cells that had just initiated cytokinesis, however, the microtubules at the anterior tip of the NFD cell did not form bundles (Fig 5G, inset at top right), and the microtubules at the nascent posterior of the OFD cell also did not form bundles (Fig 5G, inset at bottom middle). In the control cells at late stages of cytokinesis with a long cleavage furrow, the microtubules at the anterior tip of the NFD cell and at the nascent posterior of the OFD cell appeared to form bundles (Fig 5D, middle and right panels at the bottom). Note that microtubule bundling at the nascent posterior of the OFD cell occurred at the middle portion of the ventral edge of the cell (Fig 5D). In the KIN13-5 RNAi cells at late stages of cytokinesis, the microtubules at the anterior tip of the NFD cell formed bundles, but the microtubules at the nascent posterior of the OFD cell,

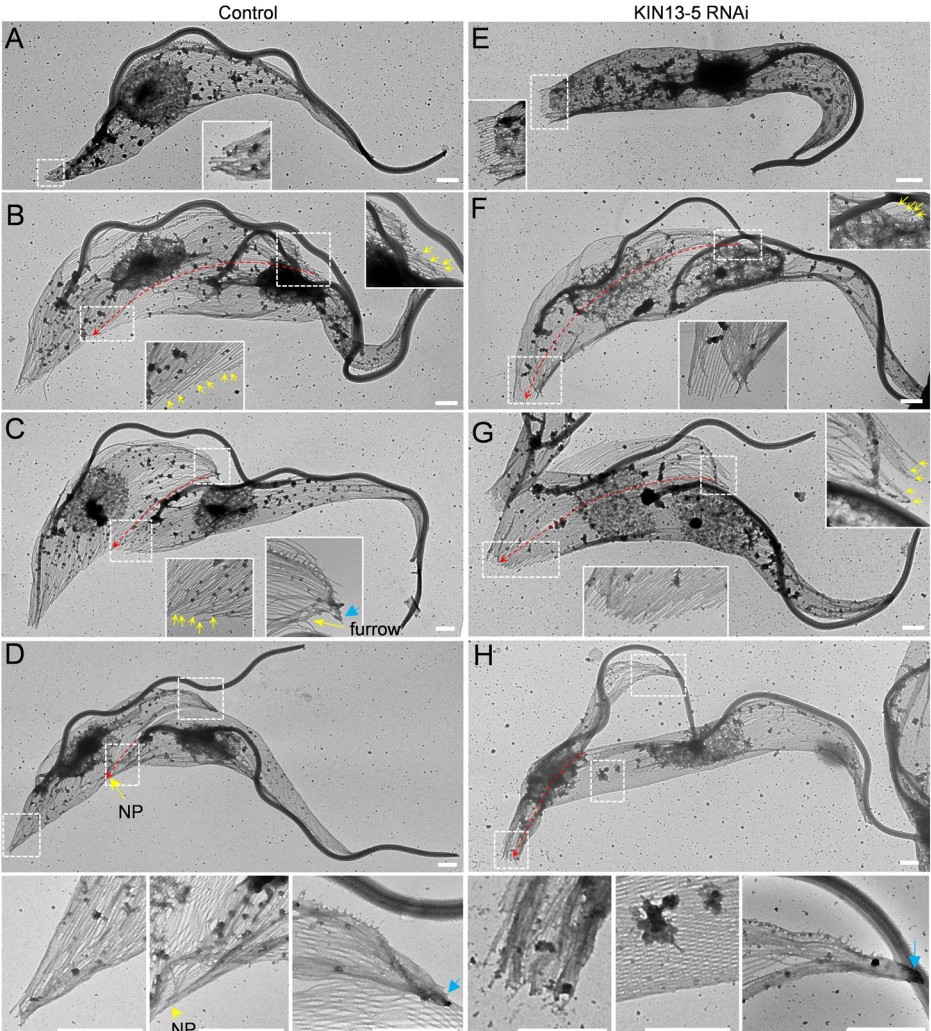

**Fig 5. Depletion of KIN13-5 by RNAi inhibits microtubule bundling at the posterior of the NFD cell and the nascent posterior of the OFD cell.** (**A-D**). Transmission electron microscopic images of a 1N1K cell (**A**), a 2N2K prior to cytokinesis initiation (**B**), a 2N2K cell at an early stage of cytokinesis (**C**), and a 2N2K cell at a late stage of cytokinesis (**D**) from the non-induced cell population. The three images under panel **D** are the zoom-in images of the selected areas in panel **D**. Scale bars: 1 μm. (**E-H**). Transmission electron microscopic images of a 1N1K cell (**E**), a 2N2K prior to cytokinesis initiation (**F**), a 2N2K cell at an early stage of cytokinesis (**G**), and a 2N2K cell at a late stage of cytokinesis (**H**) from the KIN13-5 RNAi-induced cell population. The three images under panel **H** are the zoom-in images of the selected areas in panel **H**. The yellow arrows indicate individual microtubules, the blue arrows indicate the anterior end of the NFD cell, and the red dash-line arrows indicate the cell division fold or the path of the cleavage furrow, which was determined by following the direction of extension of the subpellicular microtubules. Scale bar: 1 μm.

which was placed next to the existing posterior, did not form bundles (Fig 5H, left and right panels at the bottom). Note that the microtubules at the middle portion of the ventral edge of the cell did not form bundles (Fig 5H, middle panel at the bottom). These observations suggest that KIN13-5 knockdown impairs microtubule bundling at the anterior tip of the NFD cell during early cytokinesis stages and at the nascent posterior of the OFD cell during late cytokinesis stages.

## KIN13-5 possesses *in vitro* microtubule bundling activity

Kinesin-13 family kinesins in animals possess microtubule-depolymerizing activity to regulate microtubule dynamics during mitosis, and are characterized by the presence of a centrally located motor domain, as opposed to the other families of kinesins where the motor domain is located either at the N-terminus or the C-terminus [42]. Kinesin-13, like all kinesins, binds to microtubules with its globular head domain (motor domain), but unlike other families of kinesins, members of the kinesin-13 family contain two separate microtubule-binding sites, Kin-Tub-1 and Kin-Tub-2, within the motor domain [43, 44]. The second microtubule-binding site in kinesin-13 contains several positively charged residues and enhances tubulin cross-linking and microtubule bundling activities, independent of the microtubule-depolymerizing activity of kinesin-13 [44, 45]. The disruption in microtubule bundling at the posterior tip and the nascent posterior of *T. brucei* cells by KIN13-5 RNAi suggests that KIN13-5 may also possess microtubule bundling activity. Like other kinesin-13 family proteins, KIN13-5 also contains a centrally located motor domain (Fig 6A). It contains a sterile alpha motif (SAM) at the N-terminus and two short coiled-coil (CC) motifs at the C-terminus (Fig 6A and 6B). The SAM and the CC motif are likely involved in forming homo- and hetero-multimers [46].

To biochemically characterize the activity of KIN13-5, we expressed and purified hexahistidine-tagged KIN13-5, KIN13-5 SAM-deletion mutant, KIN13-5 CC-deletion mutant, and the motor domain of KIN13-5, as well as KIN13-1, KIN-G, and TbSpef1 to serve as controls, from bacteria to near homogeneity (Fig 6C). We first tested whether KIN13-5, KIN13-5 deletion mutants, and the motor domain of KIN13-5 were able to form multimers by blue native gel electrophoresis, and the results showed that KIN13-5 and its mutants formed distinct multimers (Fig 6D). The full-length KIN13-5 was detected as a slow-mobility band between 480kDa and 720kDa, which is likely an octamer (8-mer), and a fast-mobility band above 66kDa, which is apparently a monomer (Fig 6D). The SAM domain-deletion mutant of KIN13-5 was detected as a distinct band around 242kDa (Fig 6D), suggesting the formation of a tetramer. The CC motif-deletion mutant of KIN13-5 was detected as a distinct band between 66kDa and 146kDa (Fig 6D), suggesting the formation of a dimer. It appears that the N-terminal SAM domain is required for KIN13-5 dimerization, whereas the C-terminal CC motifs are required for KIN13-5 tetramerization. The motor domain (MD) of KIN13-5 was detected as a band at ~146kDa, which is likely a tetramer, and a band below 66kDa, which is a monomer (Fig 6D). We also tested the potential multimerization capability for the control proteins (Fig 6E). KIN13-1 appeared to form a dimer, whereas KIN-G appeared to form a tetramer (Fig 6E). The GST-fused TbSpef1 appeared to form a tetramer (Fig 6E); however, since GST itself can dimerize (Fig 6E), it suggests that TbSpef1 probably forms a dimer. This is in agreement with the human Spef1 protein, which is known to dimerize through its C-terminal CC motif [47].

Using the purified and ultracentrifugation-clarified recombinant KIN13-5 and its various mutants, we tested whether KIN13-5 was capable of promoting microtubule bundling *in vitro*. Rhodamine-labeled tubulins were polymerized into microtubules, incubated with purified recombinant KIN13-5 or KIN13-5 mutants, and then visualized under a fluorescence microscope. We found that full-length KIN13-5 and the SAM domain-deletion mutant, KIN13-5-ΔSAM, were both capable of promoting microtubule bundling *in vitro* (Fig 7A), albeit the microtubule bundles promoted by KIN13-5-ΔSAM appeared to be shorter and bundling appeared to need longer incubation times (Fig 7A), suggesting that KIN13-5-ΔSAM might have weaker activity than the full-length KIN13-5. The CC motif-deletion mutant, KIN13-5-ΔCC, did not promote microtubule bundling at the concentration of 50 nM, but showed microtubule bundling activity at the concentration of 100 nM (Fig 7A), suggesting that

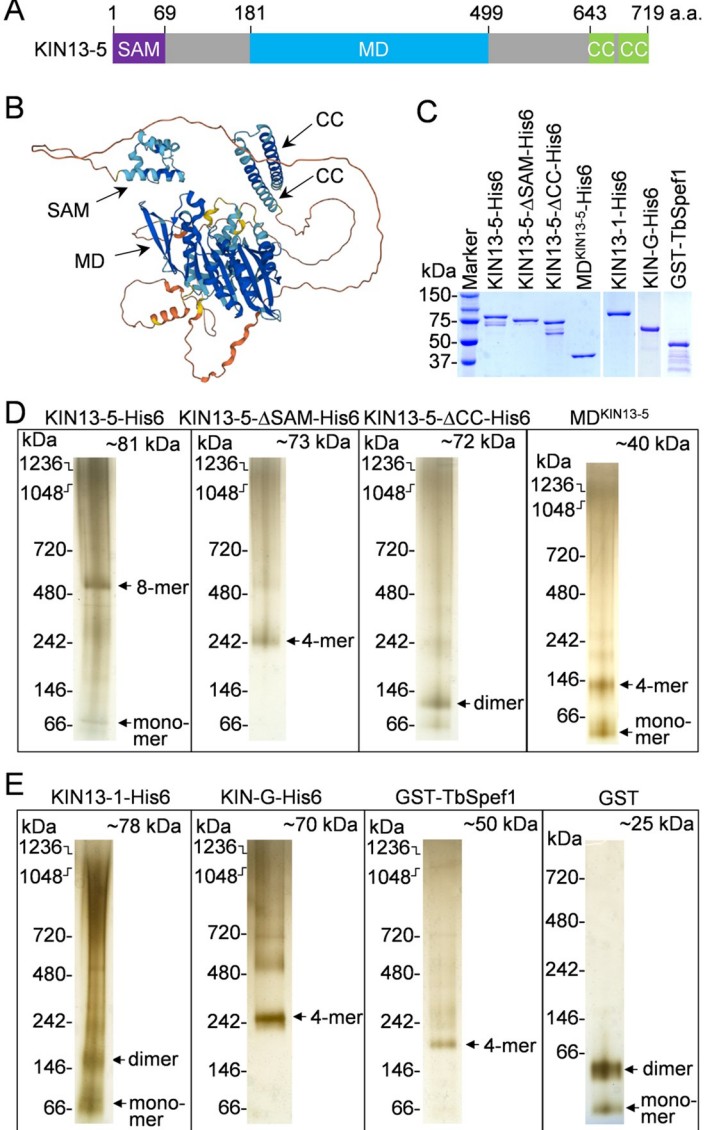

**Fig 6. KIN13-5 forms a multimer on blue native gels.** (**A**). Schematic representation of the structural domains of KIN13-5. SAM: sterile alpha motif; MD, motor domain; CC, coiled coil. (**B**). Structure of KIN13-5 predicted by AlphaFold. The different structural motifs are indicated. (**C**). Purification of recombinant hexahistidine-tagged KIN13-5, KIN13-5 mutants, the motor domain of KIN13-5, KIN13-1, KIN-G, and GST-tagged TbSpef1. (**D, E**). Analysis of purified recombinant wild-type and mutants KIN13-5 proteins (**D**) and other proteins (**E**) by blue native gel electrophoresis. Recombinant 6xHis-tagged wild-type and mutant KIN13-5 proteins, KIN13-1, KIN-G, and GST-tagged TbSpef1 were purified from *E. coli*, clarified by ultracentrifugation, and loaded onto blue native gels.

deletion of the CC motifs reduced the activity of KIN13-5. The motor domain of KIN13-5, however, was not able to bundle microtubules *in vitro* (Fig 7A), suggesting that the microtubule bundling activity of KIN13-5 does not require its motor activity.

As controls, we also tested the activity of KIN13-1, which was previously demonstrated to depolymerize microtubules *in vitro* [33], KIN-G, which has unknown activity, and TbSpef1, which was previously demonstrated to bundle microtubules *in vitro* [48]. The microtubules incubated with KIN13-1 became shorter, and the number of microtubules was decreased (Fig

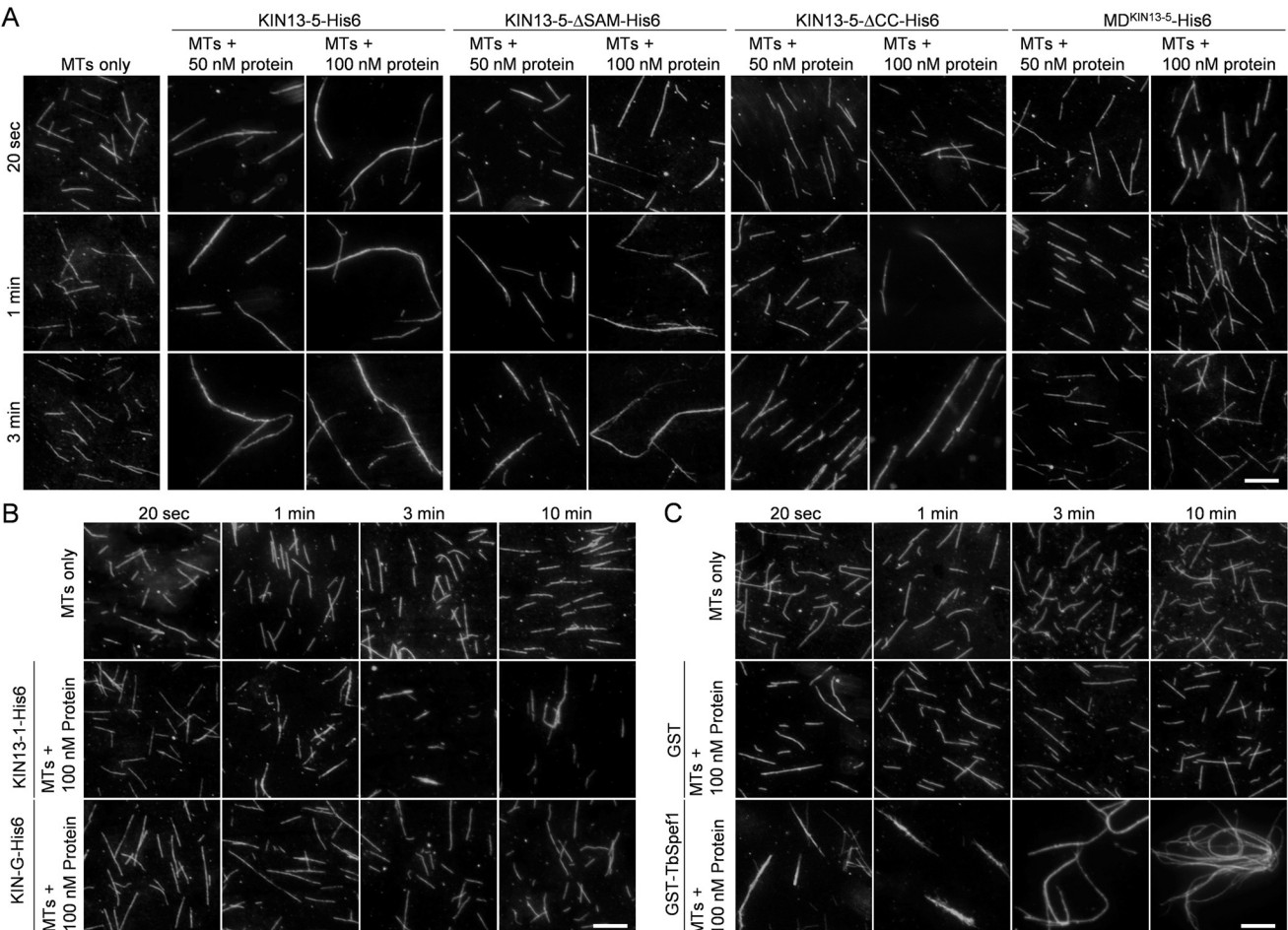

**Fig 7. KIN13-5 bundles microtubules *in vitro*.** (**A**). Microtubule bundling assay for wild-type and mutant KIN13-5 proteins. MT: microtubule. Scale bars: 5 μm. (**B**). Microtubule bundling assay for KIN13-1, a known microtubule depolymerase in *T. brucei*, and KIN-G, an orphan kinesin with unknown functions in *T. brucei*. MTs: microtubules. Scale bars: 5 μm. (**C**). Microtubule bundling assay for GST-fused TbSpef1, a known microtubule-binding protein with microtubule bundling activity. GST served as a control for GST-TbSpef1. MT: microtubule. Scale bars: 5 μm. In all panels, the recombinant proteins were purified from *E. coli* and then clarified by ultracentrifugation to remove protein aggregates before the assay.

7B, 3 min and 10 min), confirming that KIN13-1 possesses *in vitro* microtubule-depolymerizing activity. The orphan kinesin KIN-G did not show any detectable microtubule bundling or depolymerizing activity (Fig 7B). TbSpef1 was found to promote microtubule bundling (Fig 7C), consistent with the previous report [48].

## KIN13-5 acts downstream of FPRC in the cytokinesis regulatory pathway

The interaction of KIN13-5 with FPRC (Fig 1B) indicates that KIN13-5 may function in the FPRC-mediated cytokinesis pathway in *T. brucei*. To test this possibility, we first examined the potential effect of FPRC knockdown on the subcellular localization of KIN13-5. KIN13-5 was endogenously tagged with a C-terminal triple HA epitope and FPRC was endogenously tagged with a C-terminal PTP epitope in the FPRC RNAi cell line, and RNAi was induced to knockdown FPRC. Immunofluorescence microscopy demonstrated the depletion of FPRC in cells (Fig 8A), and in these FPRC knockdown cells the KIN13-5 fluorescence signal at the new FAZ tip was either reduced (~53% of the 2N2K cells) or lost (~39% of the 2N2K cells) (Fig 8A and

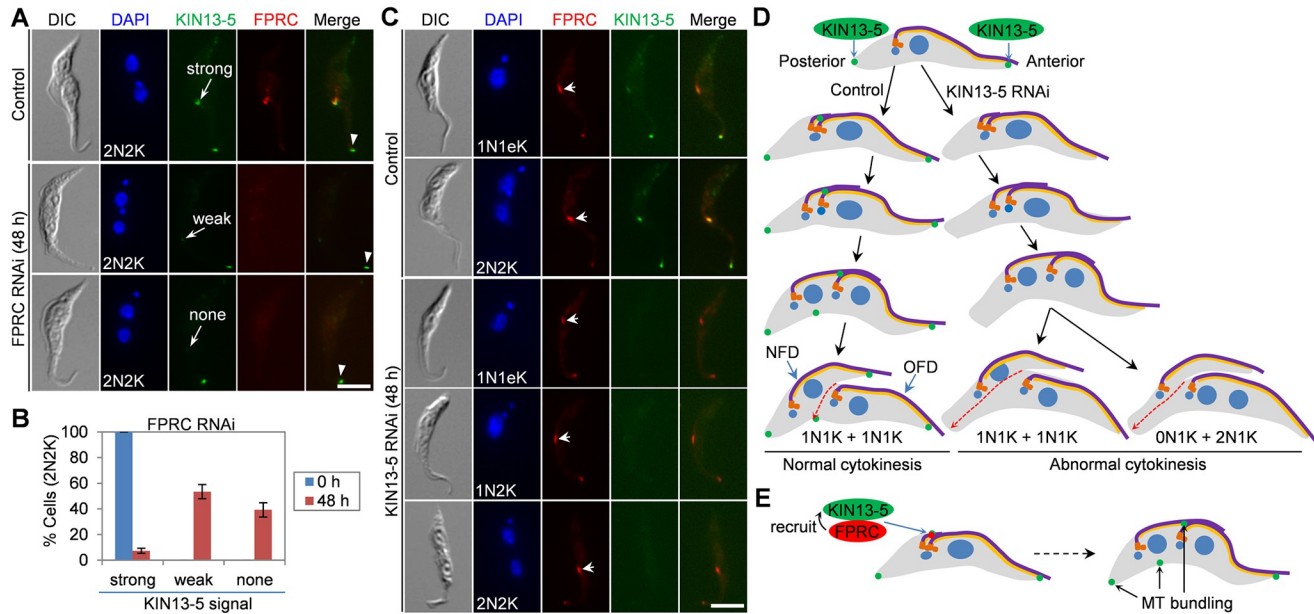

**Fig 8. KIN13-5 functions downstream of FPRC in the cytokinesis regulatory pathway.** (**A**). Effect of FPRC knockdown on the localization of KIN13-5 to the new FAZ tip. KIN13-5 and FPRC were endogenously tagged with a C-terminal triple HA epitope and a C-terminal PTP epitope, respectively, and detected with FITC-conjugated anti-HA antibody and anti-Protein A antibody. Arrowheads indicate KIN13-5 signal at the old FAZ tip which is not affected by FPRC knockdown. Scale bar: 5 μm. (**B**). Quantitation of cells with different KIN13-5 fluorescence intensity before and after FPRC RNAi. Error bars indicate S.D. from three independent experimental replicates. (**C**). Effect of KIN13-5 RNAi on the localization of FPRC. KIN13-5 and FPRC were endogenously tagged with a C-terminal triple HA epitope and a C-terminal PTP epitope, respectively, and detected with FITC-conjugated anti-HA antibody and anti-Protein A antibody. Arrows indicate FPRC at the new FAZ tip. Scale bar: 5 μm. (**D**). Summary of the effect of KIN13-5 RNAi on cytokinesis. (**E**). Model of KIN13-5 function in microtubule bundling and the regulation of KIN13-5 by FPRC.

8B), albeit the KIN13-5 fluorescence signal at the old FAZ tip was unaffected (Fig 8A). These results suggest that knockdown of FPRC impaired the localization of KIN13-5 to the new FAZ tip. Conversely, we investigated the potential effect of KIN13-5 knockdown on the localization of FPRC. KIN13-5 and FPRC were endogenously tagged with a C-terminal triple HA epitope and a C-terminal PTP epitope, respectively, in the KIN13-5 RNAi cell line. Immunofluorescence microscopy showed the depletion of KIN13-5 in cells after RNAi induction for 48 hours (Fig 8C), but in these KIN13-5 knockdown cells FPRC was still detectable at the new FAZ tip in all the cells examined (Fig 8C), demonstrating that KIN13-5 knockdown did not affect FPRC localization. Together, these results suggest that KIN13-5 functions downstream of FPRC in the FPRC-mediated cytokinesis regulatory pathway.

## Discussion

The early branching eukaryote *T. brucei* uses an actomyosin-independent machinery and a trypanosome-specific signaling pathway to control cytokinesis, the final step of cell division. The cytokinesis regulatory pathway in *T. brucei* is composed of several evolutionarily conserved proteins, such as the Polo-like kinase homolog TbPLK [13,14], the Aurora B kinase homolog TbAUK1 [11,12], and the microtubule-severing enzymes Katanin60 and Katanin80 [17,49], and a cohort of trypanosome-specific regulators, including CIF1-CIF4, FPRC, KPP1, and KLIF [16–22]. All of these cytokinesis regulators localize to the new FAZ tip, which is equivalent to the anterior tip of the NFD cell and is considered the cytokinesis initiation site, and some of them localize to the ingressing cleavage furrow during cytokinesis. In this report

we identified another new FAZ tip-localized cytokinesis regulator, KIN13-5, which functions in the FPRC-mediated cytokinesis regulatory pathway in procyclic trypanosomes. In addition to localizing to the new FAZ tip, KIN13-5 also localizes to the old FAZ tip, the posterior cell tip, the nascent posterior of the OFD cell, and the basal body (Fig 1). The localization of KIN13-5 at multiple subcellular structures suggests roles for KIN13-5 at these structures, and our functional characterization demonstrated that KIN13-5 played essential roles at some of these subcellular structures, such as the new FAZ tip, the posterior cell tip, and the nascent posterior of the OFD cell, where KIN13-5 promotes cytoskeletal microtubule bundling (Figs 2–5). It should be noted that the KIN13-5 fluorescence signal at the basal body was weaker than that at other subcellular structures and sometimes was barely detectable (Fig 1). RNAi of KIN13-5 did not appear to affect basal body duplication and segregation, because there was no significant increase in 2N1K cells after KIN13-5 RNAi (Fig 2), suggesting that KIN13-5 at the basal body likely does not play any essential roles. Additionally, there were no apparent defects in the morphology of the cytoskeleton at the anterior cell tip after KIN13-5 RNAi (Figs 2–5), implying that KIN13-5 at the old FAZ tip or the anterior cell tip likely also does not play any essential roles. Rather, since KIN13-5 remained at the new FAZ tip or the anterior tip of the NFD cell after cytokinesis was initiated (Fig 1), it is possible that KIN13-5 would remain there after cytokinesis completion, resulting in the constant localization of KIN13-5 at the old FAZ tip, where it likely does not play any essential function (see Fig 8E below). Nonetheless, we have identified KIN13-5 as a new regulator of cytokinesis, functioning downstream of FPRC in the cytokinesis regulatory pathway.

Knockdown of KIN13-5 by RNAi caused asymmetrical cell division by misplacing the cell division plane and the cleavage furrow (Figs 2 and 3). Trypanosome cells are known to divide along a pre-formed cell division fold along the longitudinal cell axis [8], and the cell division plane appears to be determined by the length of the new flagellum and/or the new FAZ [6,7]. However, neither the length of the new flagellum nor the length of the new FAZ of the KIN13-5 RNAi cells appeared to be significantly shorter than that of the control cells (Figs 2 and 3). It suggests that the mis-placement of the cell division plane and cleavage furrow in KIN13-5 RNAi cells was not attributed to defective flagellum elongation or FAZ elongation. Rather, the NFD cell of the unequally dividing cell from KIN13-5 RNAi population apparently had a smaller cell body width, compared with the NFD cell of the equally dividing cell from the control population (Figs 2 and 3). Because KIN13-5 bundles microtubules (Fig 7 and see the discussion below), we propose that KIN13-5 bundles a definite number of microtubules at the anterior tip of the NFD cell prior to cytokinesis initiation, thereby determining the start point of the ingressing cleavage furrow to ensure symmetrical cytokinesis (Fig 8D). In this regard, when KIN13-5 is knocked down, the microtubules at the anterior of the NFD cell are not bundled (Fig 5G); thus, cleavage furrow ingression starts randomly, causing the formation of the anterior that contains varying numbers of microtubules, which leads to asymmetrical cytokinesis and produces an NFD cell with varying cell sizes (Fig 2). Additionally, because KIN13-5 also bundles microtubules at the nascent posterior of the OFD cell to determine the end point of the cleavage furrow (Fig 5C and 5D), knockdown of KIN13-5 appeared to cause the shift of the end point of the cleavage furrow toward the existing posterior of the NFD cell (Fig 5G and 5H); thus, this caused the mis-placement of the cell division plane and the cleavage furrow, leading to asymmetrical cytokinesis (Fig 8D). It should be noted that after cytokinesis initiation in KIN13-5 RNAi cells, the microtubules at the anterior tip of the NFD cell were bundled together (Fig 5H), suggesting that once cytokinesis is initiated, bundling microtubules at the anterior tip no longer requires KIN13-5.

Knockdown of KIN13-5 disrupted microtubule bundling and posterior morphology, leading to the formation of a blunt posterior end (Figs 4 and 5), suggesting that KIN13-5 also

bundles the plus-ends of microtubules at the posterior cell tip for the formation of a tapered posterior. During trypanosome cell cycle, cytoskeletal microtubules extend at the posterior tip, and this process appears to be coordinated with the cell cycle [39] and is regulated by cell cycle regulatory proteins, including the S-phase cyclin-dependent kinase CRK2 [30]. Unlike KIN13-5, however, CRK2 phosphorylates β-tubulin to inhibit microtubule polymerization; thus, knockdown of CRK2 leads to elongation and branching of the cell posterior, and sometimes leads to a blunt posterior [30]. Maintenance of a tapered posterior tip in *T. brucei* also depends on microtubule-binding proteins PAVE1 and PAVE2, which bind and stabilize subpellicular microtubules at the cell posterior [31]. It appears that maintaining a tapered posterior in procyclic trypanosomes requires different regulatory proteins of diverse functions.

A surprising finding is that as a member of the kinesin-13 family, KIN13-5 does not depolymerize microtubules, but instead it possesses microtubule bundling activity in *in vitro* assays (Fig 7). Structurally, KIN13-5 resembles other kinesin-13 family proteins characterized in animals, such as human KIF2s [50], and *T. brucei*, such as KIN13-1 [32,33]. Kinesin-13 family kinesins share similar structural organizations, with an N-terminal globular domain, followed by a positively charged neck domain composed of a coiled-coil motif, a centrally located motor domain containing ATP- and microtubule-binding motifs, and C-terminal coiled-coil motifs [50]. Within the motor domain, some kinesin-13 family proteins contain a second microtubule-binding motif, which is required for microtubule bundling [44,45]. Unlike human KIF2s (KIF2A-KIF2C), however, KIN13-5 contains an N-terminal SAM domain, which is replaced by a tudor domain in KIF2A, and the neck domain in KIN13-5 lacks most of the positively charged residues and does not form a coiled-coil structure as in KIF2A (S1B–S1D Fig). Kinesin-13 family kinesins passively diffuse to both the plus- and the minus-ends of microtubules to depolymerize microtubules from both ends [51], and this depolymerizing activity depends on the positively charged residues in the neck domain [52]. It remains unclear why KIN13-5 does not have microtubule-depolymerizing activity, but instead has a strong microtubule-bundling activity. One possibility could be due to the lack of the positively charged residues in the neck domain that increases the microtubule-depolymerizing activity, as demonstrated in human MCAK [52]. Another possibility is that KIN13-5 is a functionally diverged kinesin-13 family protein that has lost microtubule-depolymerizing activity, but gained microtubule-bundling activity during evolution. The motor domain of KIN13-5 contains the second microtubule-binding domain (S1E Fig), which was demonstrated to be required for microtubule bundling in the *Drosophila* kinesin-13 family protein KLP10A [44,45]. Finally, the microtubule-depolymerizing activity generalized for the kinesin-13 family is based on the functional characterization of multiple kinesin-13 family proteins [51], but whether this function applies to all kinesin-13 family proteins remains to be verified, and at least in the case of KIN13-5, it suggests that the kinesin-13 family kinesins do not necessarily have to possess microtubule-depolymerizing activity. Nonetheless, our *in vitro* biochemical assays (Fig 7) and genetic analysis by knocking down KIN13-5 in *T. brucei* (Figs 2–5) demonstrated an essential role for KIN13-5 in bundling microtubules.

The formation of the nascent posterior of the OFD cell during *T. brucei* cytokinesis through microtubule bundling requires two kinesin proteins, KIN13-5 (Fig 5) and KLIF [27,28]. KLIF is an orphan kinesin and localizes to the anterior tip of the NFD cell prior to cytokinesis initiation and to the ingressing cleavage furrow during cytokinesis progression [17,21]. KLIF crosslinks microtubules and stabilizes the alignment of microtubule plus-ends, thereby promoting the formation of the nascent posterior of the OFD cell for cytokinesis completion [28], but it does not appear to promote microtubule bundling at the anterior tip of the NFD cell [27], likely because KLIF is a microtubule plus end-directed kinesin. Therefore, knockdown of KLIF caused defects in cytokinesis completion, but did not impair the placement of the cell

division plane and the cleavage furrow [21,27]. Unlike KIN13-5, KLIF does not localize to the posterior cell tip; thus, it is not involved in microtubule bundling at this location [17,27]. Nascent posterior formation in *T. brucei* also requires multiple microtubule-associated proteins, including CAP50, which is required for KLIF localization to the ingressing cleavage furrow [27]. Although KIN13-5 and KLIF both possess microtubule bundling activity and their knockdowns both impair nascent posterior formation, they appear to exert their roles at different subcellular locations, with KIN13-5 functioning at the nascent posterior tip of the OFD cell and KLIF functioning at the leading edge of the ingressing cleavage furrow. Therefore, cytokinesis in KLIF knockdown cells was arrested midway before cytokinesis completion [21,27], whereas cytokinesis in KIN13-5 knockdown cells was able to reach almost the end of cytokinesis (Figs 2 and 3).

In summary, we have identified another kinesin protein named KIN13-5 that plays essential roles in cytokinesis in *T. brucei*, and we placed KIN13-5 downstream of FPRC in the cytokinesis regulatory pathway (Fig 8E). KIN13-5 bundles microtubules at the posterior cell tip, the nascent posterior of the OFD cell, and the anterior tip of the NFD cell (Fig 8E), thereby ensuring correct placement of the cleavage furrow for faithful cell division and facilitating the formation of the nascent posterior for cytokinesis completion.

## Materials and methods

### Trypanosome cell culture and RNA interference

The *T. brucei* procyclic strain 29–13, which expresses T7 RNA polymerase and tetracycline repressor [53], was cultured at 27°C in the SDM-79 medium supplemented with 10% heat-inactivated fetal bovine serum, 15 μg/ml G418, and 50 μg/ml hygromycin B. FPRC RNAi cell line was generated previously [22] and was cultured at 27°C in the SDM-79 medium containing 10% heat-activated fetal bovine serum, 15 μg/ml G418, and 50 μg/ml hygromycin B, and 2.5 μg/ml phleomycin. The *T. brucei* procyclic Lister427 strain was cultured at 27°C in the SDM-79 medium containing 10% heat-inactivated fetal bovine serum. Cells were diluted (10-fold dilution) with fresh medium containing fetal bovine serum and appropriate antibiotics every 3–4 days when the cell density reached $5 \times 10^6$ cells/ml.

To generate a KIN13-5 RNAi cell line, a 716-bp fragment of the KIN13-5 coding sequence (nt. 15–730) was cloned into the pZJM vector [54]. The resulting plasmid, pZJM-KIN13-5, was linearized by restriction digestion with NotI and used to electroporate the 29–13 strain. Transfectants were selected with 2.5 μg/ml phleomycin, and clonal cell lines were obtained by limiting dilution of the transfectants in a 96-well plate containing SDM-79 medium supplemented with 20% fetal bovine serum and three antibiotics (15 μg/ml G418, and 50 μg/ml hygromycin B, and 2.5 μg/ml phleomycin). RNAi was induced by incubating the RNAi cell line with 1.0 μg/ml tetracycline. Cell growth was monitored daily by counting the number of cells using a hemacytometer under a light microscope. Two clonal KIN13-5 RNAi cell lines were analyzed, which showed almost identical phenotypes, and only the data from the characterization of one KIN13-5 RNAi cell line were presented.

### *In situ* epitope tagging of proteins using the PCR-based method

Epitope tagging of proteins from one of their respective endogenous loci was performed by PCR-based epitope tagging method [55]. Transfectants were selected with 10 μg/ml blasticidin or 1 μg/ml puromycin and cloned by limiting dilution in a 96-well plate containing 20% fetal bovine serum and appropriate antibiotics. Specifically, KIN13-5 was tagged with a C-terminal triple HA epitope in Lister427 strain, and transfectants were selected with 1 μg/ml puromycin. For co-tagging KIN13-5 with a C-terminal triple HA epitope and FPRC with a C-terminal PTP

epitope in Lister427 strain, transfectants were selected with 10 μg/ml blasticidin in addition to 1 μg/ml puromycin. For tagging KIN13-5, KLIF, and XMAP215 in KIN13-5 RNAi cell line, KIN13-5, KLIF, and XMAP215 were each tagged with a C-terminal triple HA epitope, and transfectants were selected with 1 μg/ml puromycin in addition to 15 μg/ml G418, and 50 μg/ml hygromycin B, and 2.5 μg/ml phleomycin. For tagging FPRC in the KIN13-5 RNAi cell line containing endogenously 3HA-tagged KIN13-5, FPRC was tagged with a C-terminal PTP epitope and selected with 10 μg/ml blasticidin in addition to 15 μg/ml G418, and 50 μg/ml hygromycin B, 2.5 μg/ml phleomycin, and 1 μg/ml puromycin. Each of these transfectants was cloned by limiting dilution in a 96-well plate containing 20% fetal bovine serum and appropriate antibiotics.

## Co-immunoprecipitation and western blotting

The *T. brucei* cells co-expressing KIN13-5-3HA and FPRC-PTP were lysed by sonication in 1.0 ml immunoprecipitation buffer (25 mM Tris-HCl, pH 7.4, 100 mM NaCl, 1 mM DTT, 1.0% Nonidet P-40, and protease inhibitor cocktail). Cell lysate was cleared by centrifugation at 14,000 rpm in a microcentrifuge, and the cleared cell lysate was incubated at 4˚C with 15 μl settled IgG Sepharose beads (GE Healthcare) for 2 h. Subsequently, the sepharose beads were washed six times with the immunoprecipitation buffer (see above), and proteins bound to the beads were eluted by boiling the beads for 5 min in 1x SDS-PAGE sampling buffer. Eluted proteins were then separated by SDS-PAGE, transferred onto a PVDF membrane, and immunoblotted with anti-HA monoclonal antibody (clone HA-7, H9658, Sigma-Aldrich, 1:5,000 dilution) to detect KIN13-5-3HA and with anti-Protein A polyclonal antibody (anti-ProtA; P3775, Sigma-Aldrich, 1:5,000 dilution) to detect FPRC-PTP.

For co-immunoprecipitation of KIN13-5-3HA and CIF1, cells expressing endogenously 3HA-tagged KIN13-5 were lysed by sonication in immunoprecipitation buffer, and cleared cell lysate was incubated at 4˚C with 15 μl settled EZview™ Red anti-HA affinity gel (Sigma-Aldrich) for 1 h. Beads were then washed six times with the immunoprecipitation buffer, and proteins bound to the beads were eluted, separated by SDS-PAGE, transferred onto a PVDF membrane, and immunoblotted with anti-CIF1 antibody (1:1000 dilution) [17] to detect native CIF1 and with anti-HA antibody (clone HA-7, H9658, Sigma-Aldrich, 1:5,000 dilution) to detect KIN13-5-3HA.

## Immunofluorescence microscopy

*T. brucei* cells were washed once with PBS, adhered to the glass coverslip at room temperature for 30 min, fixed with cold methanol at -20˚C for 30 min, and finally rehydrated with PBS at room temperature for 10 min. Cells were incubated in blocking buffer (3% BSA in PBS) at room temperature for 60 min, and then incubated with either FITC-conjugated anti-HA monoclonal antibody (Clone HA-7, H7411, Sigma-Aldrich, 1:400 dilution), anti-Protein A polyclonal antibody (anti-ProtA; P3775, Sigma-Aldrich, 1:400 dilution), anti-TbSAS-6 polyclonal antibody (1:1,000 dilution) [56], anti-CC2D antibody (1:1,000 dilution) [7], or YL 1/2 anti-rat monoclonal antibody (1: 2,000 dilution) [57] at room temperature for 60 min. Subsequently, cells adhered on the glass coverslip were washed three times with PBS, and then incubated with secondary antibodies, including Cy3-conjugated anti-rabbit IgG (Sigmal-Aldrich, 1:400 dilution) and Cy3-conjugated anti-rat IgG (Sigma-Aldrich, 1:400 dilution). Cells on the glass coverslip were washed three times with PBS, air dried, and then mounted in the VectaShield mounting medium (Vector Laboratories) containing DAPI. Cells were examined using an inverted microscope (model IX71, Olympus) equipped with a cooled CCD camera (model Orca-ER, Hamamatsu) and a PlanApo N 60 x 1.42 NA lens. Images were acquired using Slide-book5 software (Intelligent Imaging Innovations, Inc.) and processed using Photoshop software.

## Scanning and transmission electron microscopy

Scanning electron microscopy (SEM) was performed using the method described in our previous publication [18]. *T. brucei* cells were fixed directly in culture media with 2.5% (v/v) glutaraldehyde for 2 h in dark at room temperature. Cells were washed three times with PBS, settled onto a glass coverslip for 60 min. Subsequently, cells were washed with deionized water for two times and then dehydrated with a series dilution of alcohol solution (30%, 50%, 70%, 90%, and 100%) at room temperature for 5 min each. Dehydrated cells were dried by critical point drying. Cells on the glass coverslips were coated with a 5-nm metal film (Pt:Pd 80:20, Ted Pella Inc.) using a sputter coater (Cressington Sputter Coated 208 HR, Ted Pella Inc.). Cells were examined using Nova NanoSEM 230 (FEI) with the parameters used as follows, 5 mm for the scanning work distance and 8 kV for the accelerating high voltage. The acquired images were processed with Photoshop software.

Whole-mount cytoskeleton of trypanosome cells was prepared according to published procedures [25,58]. Trypanosome cells were washed twice with PBS, settled onto freshly charged carbon- and Formvar-coated grids, and incubated with PEME buffer containing 1% Nonidet P-40 for 5 min. Cells on the grids were fixed with 2.5% (v/v) glutaradehyde for 20 min, washed with water three times, stained with 0.1% aurothioglucose for 20 sec, and imaged using a JEOL 1400 TEM equipped with a Gatan CCD camera at 120 kV.

## Expression and purification of recombinant proteins

To express and purify C-terminally hexahistidine-tagged KIN13-5, KIN13-5 truncation mutants, KIN-G, and KIN13-1, the DNA sequences encoding KIN13-5 (a.a. 1–719), the SAM domain-deletion mutant of KIN13-5 (a.a. 71–719), the CC-deletion mutant of KIN13-5 (a.a. 1–642), the motor domain of KIN13-5 (a.a. 170–536), KIN13-1, and KIN-G were each cloned into the NdeI and XhoI sites of the expression vector pET26b. To express and purify N-terminally GST-tagged TbSpef1, the DNA sequence encoding the full-length TbSpef1 was cloned into the BamHI and XhoI sites of the expression vector pGEX4T-3. The resultant plasmids were each transformed into the *E. coli* BL21 (DE3) strain. Expression of recombinant proteins was induced with 0.1 mM isopropyl β-d-thio-galactopyranoside for 5 h at room temperature. Bacteria cells expressing hexahistidine-tagged proteins were lysed by sonication in His-fusion protein lysis buffer (50 mM $NaH_2PO_4$, 300 mM NaCl, 10 mM imidazole, pH 8.0), and cell lysate was cleared by centrifugation at 20,000 ×*g* for 5 min at 4˚C and then incubated with Chelating Sepharose Fast Flow (GE Healthcare) beads charged with nickel ion for 30 min at 4˚C. Beads were washed five times with 1 ml washing Buffer (50 mM $NaH_2PO_4$, 1.0 M NaCl, 120 mM imidazole, pH 8.0), and hexahistidine-tagged proteins were eluted with elution buffer (50 mM $NaH_2PO_4$, 300 mM NaCl, 250 mM imidazole, pH 8.0). Bacteria cells expressing GST-TbSpef1 was sonicated in GST-fusion protein lysis buffer (0.1% TritonX-100 in PBS) and centrifuged 20,000 ×*g* for 5 min at 4˚C, and the lysate was incubated with Glutathione Sepharose 4B beads for 30 min at 4˚C. Beads were washed five times with GST-Lysis buffer and then GST-Spef1 was eluted with GST-Elution buffer (20 mM reduced glutathione, 50 mM Tris-HCl pH 9.0, 0.1% Triton X-100, 100 mM NaCl, 1 mM DTT). Purified proteins were buffer-exchanged and concentrated with Amicon Ultra Centrifugal Filters 10K (Millipore).

## Blue native gel electrophoresis

Native gel electrophoresis was performed according to NativePAGE Novex Bis-Tris Gel System (Life Technologies), and the gel was silver-stained. 10 μl of purified recombinant protein (300 ng) in BRB80-DTT buffer (80 mM Potassium-PIPES, pH 6.8, 1.0 mM MgCl2, 1.0 mM EGTA, 1.0 mM DTT) plus 5% glycerol was mixed with 10 μl ddH$_2$O and 6.6 μl 4×

NativePAGE Sampling Buffer (50 mM Bis-Tris, 50 mM NaCl, 10% w/v glycerol, 0.001% Ponceau S, pH 7.2). Then, 20 µl (225 ng) of the above mixture was loaded onto NativePAGE Novex 3–12% Bis-Tris gel. The gel was run at 150 V constantly for 1.5 h using Anode Buffer (50 mM Bis-Tris, 50 mM Tricine, pH 6.8) and Cathode Buffer (0.002% Coomassie G-250 in Anode Buffer), fixed in Fixer (40% ethanol, 10% acetic acid) for 1 h, washed four times with ddH$_2$O for 20 min, sensitized with 0.02% sodium thiosulfate for 1 min, washed with ddH$_2$O for 20 sec three times, incubated with cold 0.1% silver nitrate solution (0.1% AgNO$_3$, 0.02% formaldehyde) for 20 min at 4˚C, washed with ddH$_2$O for 20 sec three times and for 1 min once, and developed with Developing Solution (3% sodium carbonate, 0.05% formaldehyde).

### Tubulin polymerization and microtubule bundling assay

Non-labeled porcine brain tubulin (Cytoskeleton, Inc., Cat#: T240-A80, 80 µg) was mixed with rhodamine-labeled porcine brain tubulin (Cytoskeleton, Inc., Cat#: TL590M, 20 µg) in 50 µl of BRB80-DTT buffer (80 mM Potassium-PIPES, pH 6.8, 1.0 mM MgCl$_2$, 1.0 mM EGTA, 1.0 mM DTT) supplemented with 1.0 mM Guanylyl-($\alpha$,$\beta$)-methylene-diphosphonate (GMP-CPP) and incubated for 5 min at 4˚C. The mix was clarified by centrifugation at 279,000 ×$g$ for 5 min at 4˚C in a TLA120.1 rotor in an ultracentrifuge (Beckman Coulter TL-100), and the supernatant (5 µl each, 2 mg/ml) was snap-frozen in liquid nitrogen and stored at -80˚C. To make microtubule seeds, one aliquot was 1:4 diluted with BRB80-DTT buffer (final concentration: 0.5 mg/ml) and incubated for 30 min at 37˚C. Microtubule seeds were pelleted at 353,000 ×$g$ for 5 min at 27˚C, and the pellet was resuspended by pipetting with 20 µl of BRB80-DTT buffer and stored at room temperature.

To polymerize microtubules from microtubule seeds, 20 µg of rhodamine-labeled tubulin was mixed with 5 µl of the microtubule seeds and 5 µl of BRB80-DTT buffer supplemented with 2.0 mM GTP. Microtubules were then successively assembled by incubating at 37˚C with 1 µl of 1 µM Taxol in BRB80-DTT-GTP buffer (BRB80-DTT buffer plus 1.0 mM GTP) for 20 min, 1.1 µl of 10 µM Taxol for 10 min, and 1.2 µl of 100 µM Taxol for 10 min. Assembled microtubules were diluted to 0.2 mg/ml with 10 µM Taxol in the BRB80-DTT-GTP buffer. Microtubule concentration was quantified by running a 90,000 rpm (353,000 ×$g$) pellet sample from 10 µl of polymerized microtubules in SDS-PAGE gel, and then adjusted to 500 nM with 10 µM Taxol in the BRB80-DTT-GTP buffer.

For microtubule bundling assays, microtubules were centrifuged at 1500 ×$g$ for 3 min to remove any Taxol-induced microtubule bundles. Purified recombinant proteins were centrifuged at 279,000 ×$g$ for 15 min at 4˚C in a TLA120.1 rotor in an ultracentrifuge (Beckman Coulter TL-100) to remove protein aggregates. Microtubules were incubated with purified recombinant proteins (indicated concentration) in 15 µl of 10 µM Taxol in the BRB80-DTT-GTP buffer plus 5% glycerol and 1.0 mM ATP. At the 20-sec, 1-min, 3-min, and 10-min time points, a 3 µl sample was taken and fixed by mixing it with 1.8 µl of 4% paraformaldehyde in PBS. Fixed microtubule samples were observed under a fluorescence microscope within an hour.

### Homology-based structural modeling of proteins by SWISS-MODEL and AlphaFold

Analysis of KIN13-5 structural domains was carried out using the SWISS-MODEL software (https://swissmodel.expasy.org). The predicted structure of KIN13-5 was obtained from the AlphaFold protein structure database (https://alphafold.ebi.ac.uk/) [59,60].

## Statistical analysis

Statistical analysis was performed using Chi-square test or Student's two tailed t-test. Detailed *n* values for each panel in the figures were stated in the corresponding legends. For counting of cells captured by immunofluorescence microscopy, images were randomly taken to cover all cell types, and all the cells in the captured images were counted. All experiments were performed at least three times. The numerical data used to generate graphs and histograms are included as S1 Data.

## Supporting information

**S1 Fig. Co-immunoprecipitation of KIN13-5 and CIF1, and structural comparison between KIN13-5 and kinesin-13 family proteins from other organisms.** (A). Co-immuno-precipitation to test the interaction between KIN13-5-3HA and CIF1. Immunoprecipitation (IP) was performed by incubating cell lysate with anti-HA affinity gel, and western blotting was performed to detect CIF1 and KIN13-5-3HA with anti-CIF1 antibody and anti-HA anti-body, respectively. (**B**). Schematic drawing of the structural domains in *T. brucei* KIN13-5 and human KIF2A. SAM, sterile alpha motif; MD, motor domain; CC, coiled coil. (**C**). Predicted structure of KIN13 and HsKIF2A by AlphaFold. (**D**). Alignment of the neck domain from HsKIF2A, HsMCAK, and *T. brucei* KIN13-5. The residues highlighted in red indicate the conserved positively charged residues that are required for stimulating kinesin microtubule-depolymerizing activity. (E). Alignment of the second microtubule-binding motif within the motor domain from HsKIF2A, DmKLP10A, and *T. brucei* KIN13-5. Identical and conserved residues are highlighted in red and green, respectively.
(TIF)

**S1 Data. Excel spreadsheet containing, in separate sheets, the underlying numerical data for Figs 1A, 2B, 2C, 2F, 2G, 2H, 2I, 3C, 3D, 4B and 8B.**
(XLSX)

## Acknowledgments

We are grateful to Dr. Cynthia Y. He of National University of Singapore for providing the anti-CC2D antibody and to the members of the Li lab for discussions and suggestions.

## Author Contributions

**Conceptualization:** Ziyin Li.

**Formal analysis:** Huiqing Hu, Yasuhiro Kurasawa.

**Funding acquisition:** Ziyin Li.

**Investigation:** Huiqing Hu, Yasuhiro Kurasawa, Qing Zhou.

**Methodology:** Huiqing Hu, Yasuhiro Kurasawa, Qing Zhou.

**Project administration:** Ziyin Li.

**Supervision:** Ziyin Li.

**Visualization:** Huiqing Hu, Yasuhiro Kurasawa, Qing Zhou.

**Writing – original draft:** Ziyin Li.

**Writing – review & editing:** Ziyin Li.

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
