## [Decision Letter · Decision Letter 0]

1 Nov 2023

Dear Dr Li,

Thank you very much for submitting your manuscript "A kinesin-13 family kinesin in Trypanosoma brucei regulates cytokinesis and cytoskeleton morphogenesis by promoting microtubule bundling" for consideration at PLOS Pathogens. As with all papers reviewed by the journal, your manuscript was reviewed by members of the editorial board and by several independent reviewers. In light of the reviews (below this email), we would like to invite the resubmission of a significantly-revised version that takes into account the reviewers' comments.

We cannot make any decision about publication until we have seen the revised manuscript and your response to the reviewers' comments. Your revised manuscript is also likely to be sent to reviewers for further evaluation.

Sincerely,

Cynthia Y He

Academic Editor

PLOS Pathogens

James Collins III

Section Editor

PLOS Pathogens

Kasturi Haldar

Editor-in-Chief

PLOS Pathogens

orcid.org/0000-0001-5065-158X

Michael Malim

Editor-in-Chief

PLOS Pathogens

orcid.org/0000-0002-7699-2064

Reviewer's Responses to Questions

**Part I - Summary**

Reviewer #1: In this manuscript, the authors have discovered a new molecular component of the machinery controlling cytokinesis in the protist Trypanosoma brucei. It is a kinesin of the family 13 and it turns out to contribute to positioning of the cleavage furrow. Surprisingly, it does not seem to crop microtubules as most members of this family but rather, it bundles them, what could be critical during cytoskeletal remodelling at cytokinesis. As a reminder, trypanosomes do not depolymerise their cytoskeleton during cell division, so it makes a lot of sense.

The authors use well-established tools during the investigation: BioID to identify new components of the cytokinesis pathway, localisation using tagged fusion proteins, functional genomics by RNAi and electron microscopy including whole-mount cytoskeletons. In addition to that, they produce in vitro data showing the bundling activity of the new kinesin.

This combination results in a solid and well-controlled study, with nice and clear figures and a new message, further refining our molecular knowledge of cell division in trypanosomes. It will be of interest for parasitologists and also cell biologists interested in cytokinesis.

Reviewer #2: The authors identify a largely uncharacterized Kinesin-13 family protein (KIN13-5) as an interactor of components of the trypanosome flagellar attachment zone. They localise the protein to both posterior and anterior ends of the cell and show that depletion of the protein results in defects in cytokinesis with loss of organisation of the microtubules at the posterior and to a lesser extent anterior end. They also present data from recombinant protein assays suggesting that the protein is multimeric and able to bundle microtubules. Finally, they show it’s localisation to the anterior site is dependent on FPRC.

The work adds to the list of components of the cleavage furrow in trypanosomes (including a non-processive dimeric kinesin KLIF with strong evidence for MT bundling [PMID: 36787745]), but the bigger impact would be in understanding whether this protein was acting primarily at anterior or posterior end (and if the latter, how?) and also in elucidation of the evolution of an MT-bundling kinesin from the depolymerizing Kinesin-13 family. There’s lots of work on Kinesin-13 family members, especially MCAK and KIF2A, and where known they are dimeric depolymerizing motors, so the finding of a Kinesin-13 that is non-depolymerizing but which bundles MTs would be very interesting – especially if some insight could be found on the features of the protein necessary (binding site, whether ATPase dependent, how linking MTs, etc.). However, the work doesn’t bring much data on the mode of action of KIN13-5 in cytokinesis, focusing mainly on the interaction at the furrow ingress site (anterior end) when the major defect appears to be at the posterior end. Moreover, the data on bundling do not provide sufficient evidence to support the authors’ conclusions and should in my opinion be removed or subject to major revision. Without either more information on mode-of-action or conclusive data on bundling activity I afraid I don’t think the work brings sufficient new insight to cytokinesis in trypanosomes.

**Part II – Major Issues: Key Experiments Required for Acceptance**

Reviewer #1: I have one question and one comment.

Question. Can the authors exclude the fact that KIN13-5 could have depolymerising activity? Based on the images shown at figure 5, it seems that the posterior end of knockdown cells contains almost twice as many microtubule ends (about 22, 5E) compared to the control (about 13, 5A). It is even more pronounced at Fig. 5G. I appreciate that KIN13-5 bundles microtubules in vitro (Fig. 6) but the source of these is not described in M&M. Are they from trypanosomes or from another organism? The cellular context (with MAPs for example) might be different from the in vitro situation. I don’t know how representative are these images but based on them, it remains a possibility that KIN13-5 could also restrict microtubule elongation and/or reduce microtubule length, hence facilitation the formation/progression of the cleavage furrow. I would be happy to hear the authors’ opinion on this point.

Comment. Throughout the manuscript, it is written that the kinesin KD divide ‘asymmetrically’ while control cells would divide ‘symmetrically’ and (see for example fig. 7). This is not completely correct as in normal procyclic cells, the new flagellum is shorter than the old one at cytokinesis and that the daughter cells have different morphologies. This is actually visible on several images in the paper (Fig. 3-5) and has been well reported in the literature (see for example Abeywickrema et al. Mol Mic2019). This is confusing, I understand the authors want to highlight the difference but I would strongly advise to use a different wording to describe the control situation.

Reviewer #2: 1) MT bundling activity.

There is a fundamental and important distinction between a protein that bundles MTs and a preparation that aggregates them. Lots of poorly folded or otherwise misprocessed proteins expressed in bacteria will cause MTs to aggregate, but this has no relevance to the biological roles of the properly processed proteins in cells. For this reason it is usual for material going into MT assays to be highly clarified (>200,000g) to remove aggregates and also for there to be supporting evidence to suggest the remaining protein is correctly folded, active, and/or in some other way relevant to material from the organism in question. Neither seems to be the case here. In fact, in this case the native gels strongly suggest the presence of large amounts of material of heterogeneous size/shape/stoichiometry which would be indicative of misfolded protein (Fig. 6D). The amount of these seems to correlate well with the degree of MT clustering in the microscopy data (Fig. 6E) and there’s no evidence given for MT-stimulated ATPase activity that would be at least be indicative of folding of some of the motor domains in the preparation.

In addition, the turbidity assays do not seem to be interpretable (Fig. 6F). The level of noise in each sample is substantially higher than I have seen in comparable assays looking at bundling/depolymerization (e.g. PMID: 17951709, 21723831, 31280993). The baseline for each sample jumps weirdly on addition of what should be identical preparations of MTs at time zero. The depolymerizing KIN13-1 does not enhance loss of turbidity over MTs alone (which would be expected given depolymerizing activity), whereas the ‘bundling’ MT KIN13-5 does not increase turbidity at all over time – but KIN13-5 alone does. None of these data seem to make biological sense and the data seem to bear little relationship to the observations in the MT microscopy.

In all, I don’t think these data can be considered sufficient support to make the important novel claim that KIN13-5 bundles MTs. The flow chamber imaging might be more convincing if were ATP-dependent (although in this case the suggestion is that it is motor-independent), or if it was titratable with highly clarified material, or if specific point mutants remove the activity with the same level of apparent misfolding/aggregation, etc. etc..

2) Mode of action of KIN13-5 in cells.

Knockdown of KIN13-5 produced a defect in ‘placement’ of the cytokinetic furrow, but without more data on the in vitro activity of KIN13-5, I felt there was insufficient insight added by these data. KIN13-5 localises to both anterior and posterior sites. The authors show the interaction at the anterior site might be FAZ dependent, but as far as I could tell there was no defect in cytokinesis initiation (at the anterior). Instead, the major defect seems to be formation of the posterior end. As such, the key questions seem to be the role/activity of KIN13-5 at the anterior versus posterior sites and how this translates to furrow placement. How does lack of KIN13-5 cause furrow misplacement? Is there any change in furrow initiation as a result of lack of KIN13-5 at this position? What is KIN13-5 binding at the posterior sites? Is the activity and/or localisation dependent on motor activity? Does KIN13-5 link MTs in the cell? Does it bind to the +ve of MTs as other Kinesin-13s?

I’m certainly not suggesting all of these questions need to be addressed in this manuscript, but I felt that in the absence of robust data on in vitro activity, some of these issues would need to be address to warrant publication in PLoS Path.

**Part III – Minor Issues: Editorial and Data Presentation Modifications**

Reviewer #1: Please provide the source of rhodamine-labelled tubulin (which organism, how purified)(page 11 line 16 and page 21).

Reviewer #2: It is absolutely essential to interpretation of the data in Fig 6 to know the ratio of tubulin to recombinant protein. I could find no information on the amount of recombinant preparation used in the turbidity assays. The work also talks about concentrations of microtubules, but no information is given on distribution of MT lengths as would be required to calculate this, and the amount stated equates to ~640mg/ml protein for 1µm MTs. I assume concentration of tubulin is meant not MTs?

Pseudocoloring of the MT microscopy (Fig.6E) to red on black when there is only one channel presented only serves to make the MTs very difficult to see. Please change to grayscale (ideally inverted).

PLOS authors have the option to publish the peer review history of their article (what does this mean?). If published, this will include your full peer review and any attached files.

Reviewer #1: **Yes: **Philippe Bastin, Institut Pasteur

Reviewer #2: No
---

## [Decision Letter · Decision Letter 1]

26 Jan 2024

Dear  Ziyin,

We are pleased to inform you that your manuscript 'A kinesin-13 family kinesin in Trypanosoma brucei regulates cytokinesis and cytoskeleton morphogenesis by promoting microtubule bundling' has been provisionally accepted for publication in PLOS Pathogens.

Best regards,

Cynthia Yingxin He

Academic Editor

PLOS Pathogens

James Collins III

Section Editor

PLOS Pathogens

Michael Malim

Editor-in-Chief

PLOS Pathogens

orcid.org/0000-0002-7699-2064

Reviewer Comments (if any, and for reference):

Reviewer's Responses to Questions

Reviewer #2: Thanks to the authors for engaging positively with my comments. In my original review, I was broadly positive about the focus of the work, but raised issues with respect to data used to support a bundling activity for the kinesin under study and felt that without clearer evidence the authors would need information on mode-of-action in the cell to bring sufficient new insight. The authors haven’t added anything in terms of mode-of-action, but have made substantial improvements to the data supporting potential bundling activity, including removing problematic turbidity data, technical improvements to the MT experiments, additional controls, and information on molar ratios. While I think it’s fair to say it looks like there’s still a proportion of misfolded protein in some preps, the additional data substantially bolster the authors’ claim and the data presented certainly suggest that this kinesin has a bundling activity (at least at reasonably high titres) that is partially dependent on a predicted coiled-coil domain. This is an interesting and important finding and I’m grateful to the authors for their additional work. In my opinion the manuscript is now suitable for publication.

All minor issues addressed in the modified manuscript.

---

## [Editor Report · Acceptance letter]

29 Jan 2024

Dear Dr. Li,

We are delighted to inform you that your manuscript, "A kinesin-13 family kinesin in *Trypanosoma brucei* regulates cytokinesis and cytoskeleton morphogenesis by promoting microtubule bundling," has been formally accepted for publication in PLOS Pathogens.

Best regards,

Michael Malim

Editor-in-Chief

PLOS Pathogens

orcid.org/0000-0002-7699-2064